# The Effects of the Pericapsular Nerve Group Block on Postoperative Pain in Patients with Hip Fracture: A Multicenter Study

**DOI:** 10.3390/diagnostics14080827

**Published:** 2024-04-17

**Authors:** Carmine Iacovazzo, Rosario Sara, Pasquale Buonanno, Maria Vargas, Antonio Coviello, Roberta Punzo, Vincenzo Maffei, Annachiara Marra

**Affiliations:** 1Department of Neurosciences, Reproductive and Odontostomatological Sciences, University of Naples “Federico II”, Via Pansini 5, 80131 Naples, Italy; pasqual3.buonanno@gmail.com (P.B.); antonio_coviello@live.it (A.C.); dottmarraannachiara@gmail.com (A.M.); 2U.O.S.C. Postoperative Intensive Care and Hyperbaric Oxygen Therapy—Election Anaesthesiological Activity in AORN “A. Cardarelli”, Via A. Cardarelli 9, 80131 Naples, Italy; rosario.sara@libero.it (R.S.); punzoroberta562@gmail.com (R.P.); vincenzo.maffei@aocardarelli.it (V.M.)

**Keywords:** hip fracture, locoregional, pain free, early mobilization, pericapsular

## Abstract

Background: An adequate early mobilization followed by an effective and pain-free rehabilitation are critical for clinical and functional recovery after hip and proximal femur fracture. A multimodal approach is always recommended so as to reduce the administered dose of analgesics, drug interactions, and possible side effects. Peripheral nerve blocks should always be considered in addition to spinal or general anesthesia to prolong postoperative analgesia. The pericapsular nerve group (PENG) block appears to be a less invasive and more effective analgesia technique compared to other methods. Methods: We conducted multicenter retrospective clinical research, including 98 patients with proximal femur fracture undergoing osteosynthesis surgery within 48 h of occurrence of the fracture. Thirty minutes before performing spinal anesthesia, 49 patients underwent a femoral nerve (FN) block plus a lateral femoral cutaneous nerve (LCFN) block, and the other 49 patients received a PENG block. A non-parametric Wilcoxon–Mann–Whitney (α = 0.05) test was performed to evaluate the difference in resting and dynamic numerical rating scale (NRS) at 30 min, 6 h, 12 h, and 24 h. Results: the PENG block administration was more effective in reducing pain intensity compared to the FN block in association with the LFCN block, as seen in the resting and dynamic NRS at thirty minutes and 12 h follow-up. Conclusion: the PENG block was more effective in reducing pain intensity than the femoral nerve block associated with the lateral femoral cutaneous nerve block in patients with proximal femur fracture undergoing to osteosynthesis.

## 1. Introduction

Hip fracture is the most common major injury in the elderly and an important cause of mortality and morbidity. Thirty-day mortality is 10%, and 20% to 30% of patients die within one year after surgery. In Italy, about 120,000 hip fractures occur every year, with a mortality of 5% at 30 days, and 18% after 1 year [1,2,3,4,5].

This type of fracture is most common in people 80 years of age and older. For most patients, optimum treatment requires surgical management of the hip fracture.

Conservative treatment requires prolonged bedrest, which can be associated with the occurrence of numerous complications; therefore, it should be considered only in patients with serious contraindications to surgery [6].

Perioperative management of these fractures among the elderly is complicated by pre-existing comorbidities, dementia, and frailty. It is generally recommended that patients with a hip fracture undergo surgery within 48 h of hospital admission [7]. The need to optimize chronic conditions must be carefully considered, and the benefits of anticipating surgery must be balanced with the risks of undergoing surgery in less-than-optimal medical conditions [8,9,10]. A delay in surgery increases the length of hospital stay, the incidence of complications (decubitus lesions, pneumonia, thromboembolic events), and mortality [11].

A long-lasting impairment of physical functions and limitations in daily life occur in 40–60% of the survivors of surgery. Longer time between surgery and rehabilitation seems to be one of the causes [12]. Postoperative delirium frequently occurs in older people after hip fracture surgery and is associated with preventable increased hospital length of stay, morbidity, and mortality.

The main aspects of anesthetic management for hip fracture surgery that are associated with differences in outcome are analgesia, fluid resuscitation, communication within multidisciplinary pre-operative meetings, provision of daily trauma lists that prioritize hip fracture surgery, and standardized preoperative assessment guided by codified treatment plans for common medical conditions [13].

Adequate and early mobilization followed by effective and pain-free rehabilitation and a multimodal perioperative analgesia, which minimizes the need for opioids and related adverse effects, such as delirium, are critical for clinical and functional recovery in the elderly patient population [14]. Perioperative analgesia can be guaranteed by using various techniques, such as intravenous administration of analgesic drugs, including opioids, paracetamol, or non-steroidal anti-inflammatory drugs; neuraxial analgesia [15]; blockade of peripheral nerves; and local analgesic infiltration [16,17,18,19,20,21].

Opioids are very effective pain control drugs in patients with proximal femur fracture, and they should be used with caution in the elderly due to the risk of impaired kidney function and respiratory depression [22]. Other side effects include nausea, confusion, constipation, urinary retention, and development of tolerance.

The administration of morphine parenterally provides effective and rapid analgesia but is burdened by non-negligible side effects, even in the most advantageous mode of administration in PCA (patient-controlled intravenous analgesia). Elderly patients and those with impaired kidney function need a lower dosage. Subcutaneous administration is performed in very few cases in elderly patients [23].

Tramadol, an atypical central opioid analgesic, also needs a modified dosage in case of liver or kidney failure and is associated with a higher risk of delirium and seizures [24].

Paracetamol is widely used, and due to its safety profile it is the first choice of analgesic drugs in the elderly. It has few side effects beyond dose-dependent hepatocyte necrosis, which may occur in the case of overdose or liver failure. The maximum intravenous dose is 1 g every 6 h, with dose reduction in selected cases; in patients with proximal femoral fracture, scheduled paracetamol administration should continue throughout the perioperative period [17].

Non-steroidal anti-inflammatory drugs (NSAIDs) provide excellent analgesia in hip fracture patients. Non-selective or selective COX2 drugs have some significant side effects, such as an increased risk of cardiovascular and kidney damage and an increased risk of gastrointestinal bleeding. The risk of these side effects increases with age. It also appears that these drugs may increase the risk of non-union fractures. NSAIDs should be used with extreme caution in patients with hip fractures, and they are contraindicated in patients with renal dysfunction [18].

Postoperative epidural has been shown to be superior to intravenous PCA with morphine [25].

Intrathecal morphine has good analgesic efficacy but involves dose-related side effects such as hypotension, urinary retention, and respiratory depression [26].

Peripheral nerve blocks have an analgesic efficacy equal to that of the epidural technique [27]; the analgesic effect can last longer in relation to the anesthetic agent used, and this facilitates rehabilitation [12]. Various approaches exist for pain control through a peripheral nerve block.

Femoral nerve (FN) blocks administered using the landmark technique, peripheral nerve stimulation (PNS), or ultrasonography (USG) to identify the nerve and surround it with local anesthetic provide effective analgesia [27].

A greater expanding of pain management techniques has been the subject of the major review, which we invite you to read now.

Blocking of the iliac fascia (IFB) can be performed using either a loss of resistance technique or an ultrasound (US)-guided technique to block the FN, ON, and lateral femoral cutaneous nerve (LFCN) with a single injection.

The three main nerves from the lumbar plexus may be blocked by injection of local anesthetic into the facial envelope of the femoral nerve (“three-in-one block”). The femoral nerve may be localized by obtaining paresthesia, by employing a nerve stimulator, or by the loss of resistance technique. The LFCN block is a predominantly sensitive technique that has been shown to be effective in reducing pain after hip surgery.

A recent systematic review and meta-analysis showed that the FN block was more effective for postoperative pain control in hip fracture patients than intravenous administration of fentanyl [28]. It also reduced the use of additional analgesic drugs, the risk of systemic complications, and the time needed to perform spinal anesthesia to facilitate patient placement [29]. The mean NRS score was 3.25 during positioning for spinal anesthesia compared to a pre-block score of 9.03 (*p* = 0.001).

Fascia iliaca compartment blocks (FICBs) help alleviate the pain of hip fractures and make the positioning of a patient for the subarachnoid anesthesia easier.

The effect size of analgesia from these blocks is only moderate [30], and the literature suggests that the obturator nerve (ON) is not covered [26,27]. A peripheral nerve block should always be considered in addition to spinal or general anesthesia, so as to prolong the period of opioid-free postoperative analgesia [31,32,33,34].

Understanding the innervation of the hip joint and the specific nerve contributions to different regions of the joint helps to guide targeted nerve block techniques to provide effective pain relief in hip surgery and related procedures.

The anterior hip capsule is innervated by the ON, accessory obturator nerve (AON), and FN, as reported by previous anatomic studies. The anterior capsule is the most richly innervated section of the joint [14], suggesting that these nerves should be the main targets for hip analgesia [26].

The highest levels of mechanoreceptors have been found in the superolateral portion of the hip capsule. Sensitive fibers and mechanoreceptors densely populate the anterior portion of the acetabular labrum and the transverse acetabular ligament. Innervation by the ON is almost always present (about 98%), and in most cases it is a single branch that arises more frequently from the main trunk of the ON; its origin can be proximal or distal to the obturator canal or within the latter. The OAN, which is rarely present, can contribute to the innervation of the joint. The FN has no direct branches to the joint but participates with branches originating from the nerve for the pectineus muscle following the medial circumflex vessels, as well as with branches coming from the nerves for the vastus lateralis, intermediate, and medial muscles and for the rectus femoris muscle; these branches follow the lateral circumflex vessels [14,35].

Short et al. have confirmed that branches of the FN (in 100% of cases) as well as to the ON (in 100% of cases) provide the innervation for the anterior hip capsule, to which the OAN contributes in about 53% of cases. In addition, this study has determined the relationship between capsular branches and the bony landmarks [36]. The high articular branches from the FN and the OAN are consistently found between the anterior inferior iliac spine (AIIS) and the iliopubic eminence (IPE), whereas the ON is located close to the inferomedial acetabulum. Using this information, Girón-Arango et al. have developed an ultrasound-guided technique for blocking these branches to the hip, the PENG (pericapsular nerve group) block [14].

Given that satisfactory pain relief promotes improved rehabilitation, we sought to evaluate the effectiveness of the PENG block as a postoperative analgesic technique versus the combined FN and LFCN blocks in patients with proximal femur fracture undergoing osteosynthesis.

## 2. Materials and Methods

We conducted a retrospective multicenter study. After obtaining approval from the local ethics committee (Protocol number 00032575), we screened the medical records of patients with proximal femur fracture who had been admitted to the AOU Federico II and AORN A.Cardarelli (Naples, Italy) from January 2022 to October 2023. The requirement for written informed consent was waived by the same ethics committee.

We included patients with an American Society of Anesthesiologist (ASA) physical status I-III, scheduled for osteosynthesis surgery within 48 h from hospital admission. Exclusion criteria were pre-existing infection at block site; contraindication to regional anesthesia; history of opiate abuse; Mini Mental Test Score (MMTS) ≤ 6 for the difficult interpretation of pain score; and pre-existing chronic pain or cognitive dysfunction which would impede an analgesia assessment. Thirty minutes before performing subarachnoid anesthesia, patients in control group received (A) an FN block in combination with a LCFN block, and patients in case group (B) received a PENG block.

All of the patients underwent subarachnoid anesthesia, which was performed in a sitting position in the L3–L4 intervertebral space with 12 mg of 0.5% hyperbaric bupivacaine.

Pain was assessed at rest and with elevation to 15 degrees of the affected limb using the numeric rating scale (NRS) at time 0 (immediately before execution of the blocks), after 30 min (immediately before positioning for the execution of the neuraxial block), and then at 6-, 12-, and 24-h time intervals for the next 24 h [37]. Pain assessment intervals were decided in temporal predictions of the most painful surgical moments and according to the pharmacokinetics of the drugs used.

The FN blocks were performed with an ultrasound-guided technique, linear probe, and in-plane approach with a 21 G 50 mm needle by injecting 15 mL of 0.5% ropivacaine + 1% lidocaine with the addition of 4 mg dexamethasone. With the same probe, approach, and needle, the LFCN blocks were performed by injecting 5 mL of 0.5% ropivacaine. The PENG blocks were performed with a US-guided technique, convex probe, and in-plane approach; the probe was initially placed in a transverse plane above the AIIS and then aligned with the pubic ramus by rotating the probe approximately 45 degrees, thus visualizing the IPE, tendon and the iliopsoas muscle, the femoral artery, and the pectineus muscle [14]; a 21 G 50 mm needle was inserted in the latero–medial direction, positioning the tip in the muscle–fascial plane between the tendon of the psoas muscle anteriorly and the pubic branch posteriorly (Figure 1). After a negative aspiration of 20 mL, 0.5% ropivacaine + 1% lidocaine with the addition of 4 mg dexamethasone were injected as required by using internal protocols. Both techniques were performed with the patient in a supine position.

All of the patients were administered 1 g IV paracetamol every 6 h and 30 mg IV ketorolac as a rescue drug with an NRS ≥ 4.

### Statistical Analyses

With a fixed significance level of 5% and a power of 90%, the sample size was 91 patients.

Normally distributed data were compared between the study arms using the unpaired *t* test, whereas non-normally distributed data were compared using the Mann–Whitney U test. All data were reported as mean (SD) or median (25 and 75% range) as appropriate. Data were inspected and tested for distribution using a logistic regression model; none of the variables significantly contributed to the model.

The non-parametric Wilcoxon–Mann–Whitney (α = 0.05) test was performed to evaluate the difference in resting and dynamic NRS at 30 min, 6 h, 12 h, and 24 h between the PENG block and the NF block associated with the NFCL block. A *p*-value ≤ 0.05 was considered to be statistically significant.

This manuscript adheres to the applicable STROBE guidelines.

## 3. Results

The patients’ characteristics and outcomes are shown in Table 1. There were no significant differences in baseline characteristics between the groups (Table 2). Including all groups in the present study, 24 (48.98%) patients were male and 25 (51.020%) were female; the mean age of the group receiving the PENG block (group B) was 81.53 years, and the mean was 81.57 years for the control group (group A). Table 3 shows the data related to the NRS values at rest (R) and with movement (D); the highest average NRS values were found at baseline time (8.10 for the PENG group vs. 8.26 for the control group) with movement and at 12 h (1.88 vs. 2.23) after the execution of the blocks in both groups.

No adverse events directly related to block placement were reported in either group.

All the patients treated with the PENG block reported significantly reduced pain scores (Table 3) compared to the control group at each evaluation time.

Twenty-four patients (82% of patients) in group A received a rescue drug dose at least twice, meanwhile only five (17%) patients in group B received a rescue drug dose at least twice.

The Wilcoxon–Mann–Whitney result was statistically significant for the NRS value of the PENG block compared to the FN + LFCN group technique after 30 min at rest (*p* = 0.0046) and after movement (*p* = 0.00151). After 12 h of analgesic intervention, the PENG group had significantly lower NRS scores than the FN + LFCN group (*p* = 0.046). During all of the other follow-up periods, the differences were not statistically significant. Figure 2 shows a comparison box plot of the median dynamic NRS between PENG and FN + LFCN at 30 min.

A logistic regression model was established to inspect and test the distribution; none of the variables significantly contribute to the model. Thus, we can conclude that the two groups are balanced with respect to the covariates.

All of the patients treated with the pericapsular nerve group (PENG) block reported significantly reduced pain scores compared to the control group, and the highest NRS values were found at baseline time with movement and at 12 h after the execution of the blocks in both groups The Wilcoxon–Mann–Whitney test results show a superiority of the PENG block compared to the 3-in-1 technique after 30 min and after 12 h.

## 4. Discussion

Our results show that the PENG block provided excellent analgesia compared to the NF block in association with the NFCL block, both at rest and in passive movement of the hip. The patients experienced a median drop of two points in NRS pain score 30 min following the block and a median drop of one point in NRS score after 12 h.

Previous publications on the PENG block have been limited to case series and involve only small numbers of patients.

The perioperative management of hip fractures is challenging due to the multiple co-morbidities and poor physiological reserve in elderly patients. The control of pain is a pivotal component so, as part of a multimodal analgesic therapy, locoregional anesthesia plays a very important role. Since its initial use, the PENG block has created great interest among the regional anesthesia community for its analgesic benefit [14].

Adequate treatment of pain in the perioperative time supports nursing care; reduces significant complications, such as infections and thromboembolic events; and facilitates early rehabilitation.

A multimodal analgesia comprising a peripheral block may even offer greater protection of total respiratory capacity through to a potential reduction in inflammatory markers such as CRP and interleukin-6 [37,38,39].

Based on our results, the PENG block was more effective than the femoral nerve block associated with the lateral femoral cutaneous nerve block in the treatment of pain at 30 min and 12 h after execution of the blocks in patients with proximal femur fracture undergoing to osteosynthesis.

A retrospective study from 2020 has shown that, in a group of patients undergoing hip arthroplasty, the PENG block was associated with a reduction in 24 h hydromorphone consumption [40]. Similarly, Sahoo et al. have examined the analgesic efficacy of the PENG block in hip fractures and showed that the PENG block provided excellent immediate (30 min after the block) analgesia, both at rest and in passive movement of the hip [41]. A recent scoping review, involving 20 articles and 74 patients, showed that the PENG block provides sufficient analgesia or anesthesia, but the evidence is limited to case reports and case series only. As such, further studies are required to determine the effectiveness, efficacy, and safety of the PENG block [42].

Peripheral nerve blocks, compared with the intravenous administration of opioid drugs, have been shown to be more effective in reducing the use of postoperative rescue analgesia, lessening risk of systemic complications and the time required to perform spinal anesthesia, and facilitating the sitting position [17,21,43].

The PENG block has the advantage of being a pericapsular block and is able to involve the articular branches of the FN but also those of the ON and AON. It is a highly selective block that cannot lead to a reduction in quadriceps muscle strength, allowing for early rehabilitation. The FN block can block the femoral nerve and, therefore, its component destined to the hip joint can probably reach the AON, but it is not able to block the articular branches of the ON and does not present the selectivity of the PENG block.

The lateral femoral cutaneous nerve block alongside the femoral nerve block did not result in superiority in terms of analgesic efficacy, but it could be associated with the PENG block in proximal femur fracture to reduce the painful component deriving from lateral skin incision [44].

The analgesic effectiveness of the PENG block in hip fracture patients is longer than the analgesic coverage offered by the blockage of individual nerve branches [45].

Duan et al. have reported that hip fracture patients receiving a PENG block had a degree of analgesia comparable to patients undergoing to a continuous block of the iliac fascia by perineural catheter. However, the latter category had less quadriceps muscle strength than the PENG block group. Therefore, the authors have shown the greater effectiveness of the non-continuous pericapsular block (PENG) for the recovery of motility and the rehabilitation time [46].

A prospective randomized trial that enrolled 59 hip fracture patients showed that a combination of PENG block and FCIB gives greater antalgic coverage than seen in patients receiving only FCIB.

Combining the PENG block with FICB, we can target different segments of the FN and increase the analgesic effects. By administering the local anesthetic at two different locations on the same nerve, a greater percentage of nerve impulses can be blocked [47].

The PENG block offers significant advantages in the preoperative period: pain-free patient transfer, easier ideal positioning for dural puncture and spinal anesthesia, and positioning of the patient in surgery lateral decubitus increases patient comfort.

A randomized clinical trial that enrolled 60 patients showed the great efficiency of the PENG block in patient placement for subarachnoid anesthesia, and also improved comfort and analgesia of the patient in the postoperative period. These results can be explained by the spread of local anesthetic through the involvement of the OAN [48].

The PENG block was effective for pain control even in the preoperative period (in the 24 h following the trauma) as a purely antalgic strategy for hip fracture patients. Xufeng et al. have shown that this technique reduces the median NRS score by three points and lessens morphine use highlighting, the greater safety of this technique and suggesting that it could also be an elective technique for hip fracture patients planned for conservative treatment [49]^.^

To our knowledge, this is the first multicenter study on an elderly population to show that the PENG block provides effective analgesia for patients undergoing hip surgery.

In conclusion, the PENG block appears to be a technique that guarantees analgesia efficacy equal or superior to other techniques. Its safety profile makes it one of the techniques of choice in elderly populations. It increases the comfort of elderly patients with hip fracture, allowing for an early rehabilitation.

Our findings should be interpreted in the context of some limitations. First, our strict inclusion criteria limited the significance of this research to a report on a population of patients with few comorbidities. Secondly, the NRS score represents an optimal scale of pain assessment, even if it is subject to important inter-individual variability. Despite the use of data from experienced operators, in a multicentric study there is a non-modifiable variability in the execution of techniques, such technicians’ operator-dependent performance of the PENG block.

Further studies with larger samples sizes are mandatory to determine the most effective dosage and volume of local anesthetic in the PENG block. Such investigations are essential for corroborating its clinical efficacy in patients undergoing not only osteosynthesis but also various surgical procedures, such as endoprosthesis placement and arthroplasty. In addition, larger sample sizes could emphasize the impact on rehabilitation outcomes.

## Figures and Tables

**Figure 1 diagnostics-14-00827-f001:**
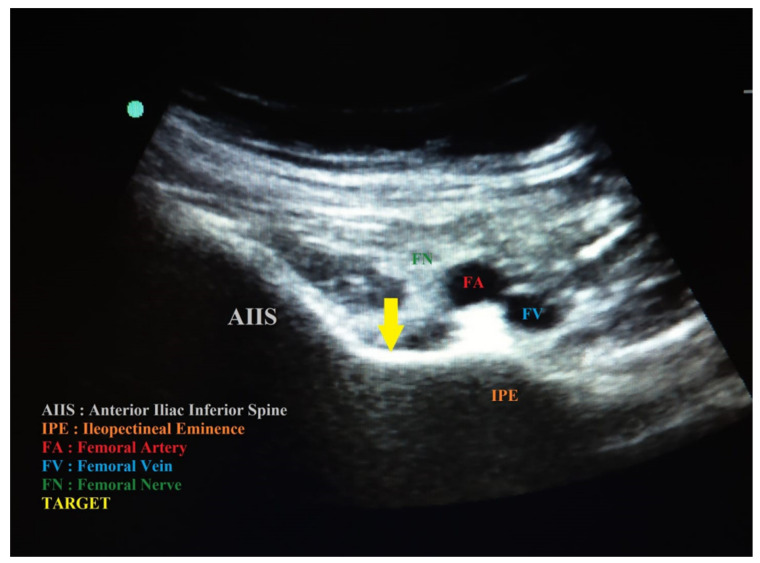
PENG block with high frequency linear probe. The needle is outlined by a yellow dotted arrow.

**Figure 2 diagnostics-14-00827-f002:**
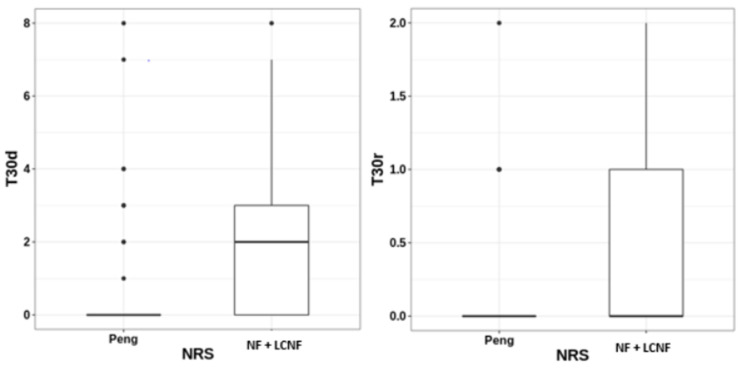
Box plot of the median of dynamic NRS between PENG and FN + LFCN at 30 min. PENG = pericapsular nerve group; NF: Nervous Femoral; LFCN: lateral femoral cutaneous nerve; PE = pertrochanteric fracture; IN = intertrochanteric fracture.

**Table 1 diagnostics-14-00827-t001:** Demographic data.

	PENG Group	NF + LFCN Group	*p* Value
	n = 49	n = 49	
**Age**	81.53 (3.44)	81.57 (4.26)	0.8962
**Fracture type**			
**PE**	25	24	0.9216
**IN**	24	25	0.9187
**Gender**			
**M**	26	25	0.9457
**F**	23	24	0.9125

PENG = pericapsular nerve group; NF: Nerve Femoral; LFCN: lateral femoral cutaneous nerve; PE = pertrochanteric fracture; IN = intertrochanteric fracture.

**Table 2 diagnostics-14-00827-t002:** Regression logistic model between groups.

Deviance Residuals:	Min	1Q	Median	3Q	Max
	−1.20170	−1.18250	0.00063	1.16556	1.20234
**Coefficients:**	**Estimate**	**Std. Error**	**z value**	**Pr (>|z|)**	
**(Intercept)**	0.155293	4.317451	0.036	0.971	
**Gender**	−0.083093	0.407533	−0.204	0.838	
**Age**	−0.001485	0.053078	−0.028	0.978	
**Type of fracture**	0.011571	0.410636	0.028	0.978	

**Table 3 diagnostics-14-00827-t003:** The NRS values at rest (r) and with movement (d) obtained at time 0, 30 min, 6 h, 12 h, and 24 h.

	**PENG**						
	**T0r**	**T0d**	**T30’r**	**T30’d**	**T6**	**T12**	**T24**
**Average score**	1.4489796	8.1020408	0.1428571	0.755102	0.6530612	1.877551	0.5714286
**Median score**	2	8	0	0	0	1	0
	**NF + LFCN (control group)**						
**Average score**	1.4489796	8.2653061	0.4693878	1.9387755	1.244898	2.244898	0.8367347
**Median score**	1	8	0	2	0	2	0
Number of Fisher Scoring iterations: 3							
**PENG vs. CTRL (Wilcoxon–Mann–Whitney test)**							
*p*-value	0.9762	0.8005	0.00457	0.001511	0.0329	0.0459	0.4434

## Data Availability

The data are available only after authorization of the Ethics Committe for ethics and privacy restriction.

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
