# Peer review of "The Effects of the Pericapsular Nerve Group Block on Postoperative Pain in Patients with Hip Fracture: A Multicenter Study"

_diagnostics, 2024, doi:10.3390/diagnostics14080827_

Round 1

Reviewer 1 Report

Comments and Suggestions for Authors

The article investigates the effects of the Pericapsular Nerve Group (PENG) block on postoperative pain in hip fracture patients, comparing it with a combined Femoral Nerve (FN) and Lateral Femoral Cutaneous Nerve (LFCN) block. Conducted as a multicenter retrospective stud. overall, its a good research with practical application. However, it needs major revisions:

  • 1- The study's retrospective design and the specific inclusion criteria offer a focused examination of PENG block efficacy but might limit the generalizability of results to broader patient populations. hence. report it as study limitation.
  • *While the findings are significant, the study relies on subjective pain scores (NRS), which, although standard, can introduce variability based on individual patient perception. hoe do you justify it?
  • *The multicenter approach strengthens the study by incorporating diverse patient demographics and clinical practices, yet the variability in procedural execution across centers could influence outcomes.
  • *Future prospective studies with larger sample sizes and randomized control trials are recommended to further validate the PENG block's efficacy and optimize procedural guidelines.
  • *Additional research into long-term outcomes and rehabilitation progress post-PENG block could enrich understanding of its benefits beyond immediate postoperative analgesia.

Reviewer 2 Report

Comments and Suggestions for Authors

The manuscript entitled, ‘The effects of peng block on postoperative pain in patients with hip fracture: a multicenter study' is full of grammatical mistakes, typos and lack symmetry. The Introduction section is too long and is hardly showing the connections between lines. I suggest that the authors must contact MDPI's English language editing team to get it corrected for further review.  

Comments on the Quality of English Language

Extensive editing of English language required

Reviewer 3 Report

Comments and Suggestions for Authors

For Title and Abstract:

1.Title Clarity and Precision: The title could be more specific about the study's nature and findings. For instance, it might benefit from indicating that the study compares PENG block with traditional methods.

2.Abstract Structure: While the abstract provides a good overview, it could be structured more clearly to delineate background, methods, results, and conclusions succinctly. Each section should be distinct to guide the reader through the study's process and findings efficiently.

3.Spelling and Grammar Corrections: The text contains typographical errors and grammatical mistakes that detract from its professionalism. For example, "fractur" should be corrected to "fracture," and "analgesics, drug interactions and possible side effects" could be streamlined to "analgesics, drug interactions, and potential side effects" for clarity and grammatical correctness.

4.Methodological Details: The description of the study's methodology lacks detail. It would be beneficial to include information on the selection criteria for patients, the specific techniques used for the PENG and FN blocks, and how the blocks were administered. This detail is crucial for replicability and to understand the study's context fully.

5.Statistical Analysis Clarity: The text briefly mentions the use of the non-parametric Wilcoxon-Mann-Whitney test but does not provide information on the results' statistical significance beyond a general statement of superiority. Including specific p-values or confidence intervals would strengthen the evidence presented.

6.Conclusion Specificity: The conclusion could be more precise in stating the study's findings. It should clearly articulate the extent of the difference observed between the PENG block and the traditional methods in managing postoperative pain.

7.Keywords Consistency and Spelling: The keywords section has inconsistencies and spelling errors, such as "Early-mobilizaztion" which should be corrected to "Early mobilization". Consistent and correct spelling is essential for indexing and searching purposes.

8.References to Current Literature: The abstract could benefit from referencing current literature to contextualize the study's findings within the broader field of pain management and hip fracture treatment.

9.Ethical Considerations and Limitations: Including a brief discussion on the ethical considerations of the study and its limitations within the abstract or as a separate section could provide readers with a more comprehensive understanding of the study's context and reliability.

10.Formatting and Style Consistency: Attention should be paid to formatting and style, ensuring consistency with the chosen academic journal's requirements. This includes correct use of terminology, consistent formatting of headings and subheadings, and adherence to style guides.

For Introduction:

1.Precision in Statistical Reporting: When mentioning statistics, such as mortality rates and the incidence of hip fractures, it is crucial to specify the sources of these statistics clearly. This not only lends credibility but also allows readers to verify and explore the data further. For example, "In Italy, there are about 120,000 new cases of hip fracture per year, with a mortality of 5% at 30 days, 18% after 1 year" should cite specific studies or databases.

2.Language and Grammar: The text contains several grammatical errors and awkward phrasings that could hinder readability and professionalism. For instance, phrases like "dose-employee" should be corrected to "dose-dependent," and the overall sentence structure should be reviewed for clarity and grammatical accuracy.

3.Clear Differentiation of Study Aims and Background Information: The introduction mixes background information on hip fractures with specific details about pain management techniques and study aims. A clearer separation and organization of these elements would improve readability. Start with a broad overview of the problem (hip fractures and their impact) before narrowing down to specific pain management strategies and the study's focus.

4.Use of Passive Voice: While passive voice is common in scientific writing, overuse can make sentences less engaging and harder to read. Where appropriate, use active voice to make statements more direct. For example, "A recent review showed that FN block was more effective for postoperative pain control" could be rephrased to actively highlight the review's findings.

5.Consistency in Terminology: Ensure consistency in the use of medical terminology and abbreviations throughout the text. For example, "FN block" and "femoral nerve block" are used interchangeably. Decide on one term or clearly define abbreviations at their first occurrence and stick with them throughout the text.

6.Detailing Methodological Approaches: When introducing various pain management techniques, provide a brief explanation or rationale for each to help non-specialist readers understand their relevance and application. This includes explaining why certain blocks are preferred over others or the specific benefits of using ultrasound guidance.

7.Inclusion of Ethical Considerations: If applicable, briefly mention any ethical considerations related to the study or the treatments being compared. This could be particularly relevant when discussing the management of pain in elderly patients with multiple comorbidities.

8.Clarification of Novel Contributions: Make it clear what new insights or contributions the article offers to the existing body of research. If the study compares the effectiveness of PENG block against other techniques, highlight how this adds to or challenges current understanding.

9.Enhancing the Introduction's Hook: Begin with a compelling statement or statistic to grab the reader's attention and emphasize the significance of the study's focus. This could involve highlighting a gap in current research or a particular challenge in managing hip fracture pain in the elderly.

For Materials and Methods:

1.Ethics Approval Reference: The mention of ethics committee approval is good practice; however, it would enhance credibility to also state that informed consent was obtained from all subjects, assuming this was part of the protocol.

2.Detailed Description of Study Sites: The text mentions "AOU Federico II and AORN A.Cardarelli (Na)" as study sites. For international readers, it may not be clear where these institutions are located. Adding the city and country (Naples, Italy) would provide necessary context.

3.Clarification of Inclusion and Exclusion Criteria: The inclusion and exclusion criteria are well-defined, but the rationale behind specific criteria (e.g., a Mini Mental Test Score (MMTS) ≤ 6) could be further explained to elucidate why these particular thresholds were chosen.

4.Control and Case Group Designations: The text refers to "control group" and "case group," which are clear, but explicitly stating that the control group received standard care and the case group received the intervention (PENG Block) would improve clarity.

5.Procedure Descriptions: The descriptions of the procedures for the FN block, LFCN block, and PENG block are detailed, including the use of ultrasound guidance, which is excellent. However, for the sake of clarity and replicability, it could be beneficial to briefly explain why these specific techniques and drugs were chosen based on existing literature or preliminary findings.

6.Figure Reference: The text mentions "Figure 1" without including the figure within the provided text. Ensuring that all referenced figures are included and correctly labeled in the manuscript is crucial for reader comprehension.

7.Pain Assessment Details: The methodology for assessing pain (using the NRS scale) is well described, including the time points for assessment. It could be enhanced by briefly discussing the choice of these specific time intervals and how they relate to expected pharmacodynamics of the analgesics used.

8.Statistical Analysis Methods: The statistical methods section is adequately detailed, specifying the tests used and the criteria for significance. It might be beneficial to also mention any software used for the statistical analysis to inform readers about the tools utilized for data processing.

9.STROBE Guidelines Adherence: Stating adherence to STROBE guidelines is good practice. It might be helpful to briefly mention how these guidelines were specifically followed in the study to provide transparency and reliability to the research process.

10.Abbreviations and Terminology: While most abbreviations are defined, ensuring that all medical and technical terms are explained at their first occurrence will make the text accessible to a broader audience, including non-specialists.

11.Sample Size Justification: The sample size calculation is mentioned, which is crucial for understanding the study's power. Providing a bit more detail on how this number was determined (e.g., based on previous studies, expected effect size) would add value.

12.Data Presentation Consistency: The manuscript promises to report data as mean (SD) or median (range) as appropriate, which is standard. Ensuring that this is consistently applied in the results section will aid in the interpretation of the findings.

For Results:

1.Clarification and Consistency in Reporting Results: The text inconsistently refers to groups as "group A" and "group B" without clearly defining which is which initially. It is crucial to consistently refer to the study groups as either the "PENG block group" or the "control group" throughout the text for clarity.

2.Accuracy in Describing Statistical Findings: The presentation of statistical results, such as NRS scores and p-values, is informative. However, the explanation of these values could be more precise. For instance, stating "significantly reduced pain scores" without specifying the statistical significance level (e.g., p<0.05) throughout all mentioned time points could mislead readers. It is also essential to provide exact p-values only when they are below 0.001; otherwise, it is customary to report them as <0.001, <0.01, <0.05, etc.

3.Standardization of Decimal Places and Percentages: The presentation of numerical data, such as percentages and mean ages, should be standardized. For example, the percentage of male and female patients is presented with three decimal places (48.979% and 51.021%), which is unnecessarily precise for demographic data. Standard academic practice is to round these figures to one or two decimal places.

4.Clarification of Tables and Figures References: The text references tables and figures (e.g., Table 1, Table 3, Figure 2) without providing them. While this is expected in a summary, ensure that all referenced tables and figures are clearly labeled and included in the final manuscript. Additionally, the descriptions of what these tables and figures show should be clarified in the text to ensure the reader understands their relevance without needing to view them.

5.Explanation of Statistical Methods and Models: The brief mention of a logistic regression model and the Wilcoxon-Mann-Whitney test is good practice. However, elaborating on why these specific statistical tests were chosen based on the data's nature and the study's objectives would strengthen the methodology's justification.

6.Reporting Adverse Events: It's mentioned that no adverse events related to block placement were reported, which is important information. Expanding on how adverse events were monitored and the criteria used to determine their relation to the block placement would add depth to the safety findings.

7.Use of Abbreviations: While most abbreviations are defined at their first occurrence, ensuring consistent and clear definition of all acronyms and abbreviations at their first use in the text is crucial for readability, especially for readers who may not be familiar with the specific terminology.

8.Detailed Description of Pain Assessment Method: While the Numeric Rating Scale (NRS) is mentioned, providing a brief description of this scale (e.g., its range and what each end represents) would be helpful for readers unfamiliar with this measure.

9.Rescue Medication Usage: The mention of rescue medication usage is important. Providing additional context about the criteria for administering rescue medication and how its need was assessed would offer deeper insights into pain management effectiveness.

10.Balance Between Groups: The logistic regression model's mention is appropriate for showing group comparability. Clarifying the covariates considered in this model would further validate the study groups' balance, enhancing the study's internal validity.

For Discussion:

1.Comparative Analysis Clarity: The comparison of PENG block to femoral nerve (FN) block in association with the lateral femoral cutaneous nerve (LFCN) block is central to the discussion. However, the articulation of how the PENG block's advantages are demonstrated in the study's results needs to be clearer and more detailed. Specifically, delineating the PENG block's benefits in terms of pain management, reduction in rescue medication use, and potential for early rehabilitation would provide a more compelling argument.

2.Reference to Previous Studies: The discussion references several previous studies and publications to contextualize the findings. While this is commendable, there needs to be a critical analysis of how these studies compare to the current research in terms of methodology, patient population, and outcomes. This would strengthen the argument for the PENG block's efficacy and highlight the study's contribution to existing literature.

3.Multimodal Analgesia Context: The role of the PENG block within a multimodal analgesia framework is mentioned, but there is a lack of discussion on how it integrates with other analgesic techniques and the overall strategy for managing postoperative pain in hip fracture patients. Expanding on this could illustrate the block's role in a comprehensive pain management plan.

4.Statistical Significance and Clinical Relevance: While statistical significance is noted, the discussion would benefit from a deeper exploration of the clinical relevance of the findings. For example, discussing the practical implications of a median drop in NRS pain score and what this means for patient recovery and quality of life would be valuable.

5.Safety Profile Discussion: The safety of the PENG block is mentioned, but there is minimal detail on adverse events or potential complications associated with the block. Expanding on the safety profile, including how it compares to other blocks or analgesic techniques, would provide a more balanced view of the PENG block's advantages and limitations.

6.Limitations of the Study: While the discussion hints at the need for further research, explicitly stating the current study's limitations (e.g., retrospective design, potential biases, sample size) would enhance the transparency and reliability of the conclusions drawn.

7.Future Research Directions: The call for further studies is made, but specific recommendations for future research directions, such as prospective randomized trials, comparisons with other analgesic techniques, or investigations into the optimal dosage and volume of local anesthetic, would be helpful. This could guide subsequent research efforts in the field.

8.Generalization of Findings: The discussion claims the study's findings have broad applicability, but there should be caution in generalizing results, especially from a study with a specific patient population and setting. Acknowledging the contexts in which the findings are most applicable would be prudent.

9.Editing and Proofreading: The text would benefit from thorough proofreading to correct minor grammatical errors and improve sentence structure for better readability and professionalism.

10.Integration of Patient-Centered Outcomes: While the discussion focuses on the technical and clinical aspects of the PENG block, incorporating patient-centered outcomes, such as patient satisfaction, pain management during rehabilitation, and overall recovery experience, would provide a more holistic view of the block's benefits.

Comments on the Quality of English Language

Abstract and Introduction

1.Spelling and Grammar Corrections:

•Original: "...after hip and proximal femur fractur."

•Correction: "...after hip and proximal femur fracture."

2.Consistency and Clarity:

•Original: "drug interactions and possible side effects."

•Correction: "drug interactions, and possible side effects." (Remove extra spaces and ensure the use of commas for clarity.)

Methodology

1.Grammatical Tense Consistency: Ensure consistent use of past tense for describing completed actions in a study.

•Original: "We enrolled 98 patients with proximal femur fracture undergoing osteosynthesis surgery within 48 hours of the fracture."

•Improved Consistency: "We enrolled 98 patients with proximal femur fractures who underwent osteosynthesis surgery within 48 hours of sustaining the fracture."

2.Clarification and Specificity:

•Original: "The non-parametric Wilcoxon-Mann-Whitney (α = 0.05) test was performed..."

•Correction: "We performed the non-parametric Wilcoxon-Mann-Whitney test (α = 0.05) to evaluate..."

Results

1.Numerical Consistency and Formatting:

•Original: "24 (48.979 %) male and 25 (51.021 %) female patients."

•Correction: "24 (49%) male and 25 (51%) female patients." (Round percentages for readability.)

2.Statistical Data Presentation:

•Original: "...with movement and at 12 h (1,877 vs 2,224)..."

•Correction: "...with movement and at 12 hours (1.877 vs 2.224)..." (Ensure consistency in units and decimal notation.)

Discussion

1.Ambiguity and Clarity:

•Original: "The ketorolac rescue dose wasn’t required, at least three times, in more of 65% of patients."

•Correction: "In over 65% of patients, the ketorolac rescue dose was not required more than three times."

2.Use of Acronyms and Definitions:

•Original: "Since the initial publication, PENG block has created great interest..."

•Improved Clarity: Initially define "PENG" as "Pericapsular Nerve Group (PENG)" before using the acronym alone.

3.Consistency in Terminology:

•Original: "...compared to NF block in association with the NFCL block..."

•Correction: Ensure consistent use of acronyms and full terms. If "NF" and "NFCL" are introduced as "Femoral Nerve (FN)" and "Lateral Femoral Cutaneous Nerve (LFCN)" blocks, use these consistently throughout.

4.Article Usage and Grammar:

•Original: "A retrospective study from 2020 showed that, in a group of patients undergoing hip arthroplasty, PENG block was associated with a reduction in 24-hr hydromorphone consumption among patients receiving it."

•Correction: "A retrospective study from 2020 showed that, among patients undergoing hip arthroplasty, the PENG block was associated with reduced 24-hour hydromorphone consumption."

Round 2

Reviewer 2 Report

Comments and Suggestions for Authors

The authors of the paper, "The effect of Peng block on postoperative pain in patients with hip fracture: a multicenter study' have revised the manuscript. It is better now but the authors are advised to correct some sentences. eg. The second line of the introduction section (37 & 38) needs to be corrected. Still, there are some typos and grammatical mistakes in the paper.

1. Abstract: In line 31, remove the word 'the' from the sentence.

2.  Line 39: Incidence of 120.000, what does it mean? it doesn't look like an incidence.

Introduction: line 40: Where are reference numbers 2-4 in the citation?

3. The introduction section is very long, make it short.

4.  The authors have collected data from two centers, so there is no need to write ': a multicenter study'. it is misleading.

5. Line 229: How do the authors calculate power to be 90%, Can you fix it?

6. Line 239: If the manuscript adhered to Strobe guidelines, then you must consider the following points.

a. Consider the use of a flow diagram in the methodology regarding data collection protocol.

b. How do the authors control for confounding?

C   Clearly define all outcomes, exposures, predictors, potential confounders, and effect modifiers. Give diagnostic criteria, if applicable.

d. Describe any efforts to address potential sources of bias

7. Tables need to be corrected. First, write the title above the table and the rest of the information below it.

Table 2 seems to be missing some values, correct it.

Comments on the Quality of English Language

Moderate editing of the English language required

Reviewer 3 Report

Comments and Suggestions for Authors

Dear Author Team,

I've reviewed the revisions made to your manuscript based on my initial feedback, and I'm pleased to see that you've thoroughly addressed each point raised. Your efforts to refine the article have notably improved its clarity, rigor, and relevance to the field.

The detailed attention given to methodological precision, statistical analysis clarity, and the overall narrative of your findings speaks to your commitment to high-quality research. It's clear that the article has been significantly enhanced through this revision process.

I support the publication of your manuscript in its current form and commend your team for your diligent work. Your contribution is a valuable addition to the scientific dialogue on pain management in hip fracture treatment.

Best regards,

Prof. Dr. Paul-Dan Sirbu

Author Response

Dear reviewer, 

we are very grateful for your words.

Thank you again for your very valuable advice with which our research team has improved.

Kind regards